# Genetic Parameters of Growth Traits and Quantitative Genetic Metrics for Selection and Conservation of Mecheri Sheep of Tamil Nadu

**DOI:** 10.3390/ani13030454

**Published:** 2023-01-28

**Authors:** Balakrishnan Balasundaram, Aranganoor Kannan Thiruvenkadan, Nagarajan Murali, Jaganadhan Muralidharan, Doraiswamy Cauveri, Sunday Olusola Peters

**Affiliations:** 1Department of Animal Genetics and Breeding, Veterinary College and Research Institute, Tamil Nadu Veterinary and Animal Sciences University, Orathanadu 614 625, India; 2Department of Animal Genetics and Breeding, Veterinary College and Research Institute, Tamil Nadu Veterinary and Animal Sciences University, Salem 636 112, India; 3Department of Animal Genetics and Breeding, Veterinary College and Research Institute, Tamil Nadu Veterinary and Animal Sciences University, Namakkal 637 002, India; 4Mecheri Sheep Research Station, Tamil Nadu Veterinary and Animal Sciences University, Salem 636 453, India; 5Department of Animal Genetics and Breeding, Madras Veterinary College, Tamil Nadu Veterinary and Animal Sciences University, Chennai 600 007, India; 6Department of Animal Science, Berry College, Mt Berry, GA 30149, USA

**Keywords:** genetic parameters, (co)variance components, Mecheri sheep, animal models, maternal genetic influence, inbreeding

## Abstract

**Simple Summary:**

Mecheri sheep are a hair-type sheep breed from India, reared for meat, which play a vital part in the economic well-being of farmers. This study was conducted to evaluate Mecheri sheep by assessing sources of variation and estimating the genetic parameters for growth traits and examining genetic diversity through pedigree analysis. The estimated direct heritability for growth traits was lower in post-weaning age groups when compared to the pre-weaning stages. The intermediate heritability and moderate-to-high genetic correlations observed for the weaning weight indicated that the weaning weight could be employed as a selection criterion for enhancing later-age traits of body-weight trait. Mecheri sheep demonstrate a much lower level of inbreeding due to the widespread use of techniques such as routine genetic diversity monitoring and ongoing efforts to scientifically design and carrying out mating plans for the flock.

**Abstract:**

Determining the genetic and non-genetic sources of variation in a breed is vital for the formulation of strategies for its conservation and improvement. The present study was aimed at estimating the (co)variance components and genetic parameters of Mecheri sheep by fitting six different animal models in the restricted maximum likelihood method, with a preliminary investigation on the performance of animals for non-genetic sources of variation. A total of 2616 lambs were studied, and varying levels of significance were found for the effects of period, season, parity of dam, and birth type on different body-weight traits. Direct heritability estimates derived from the best animal model for body weight at birth, three months, six months, nine months, and twelve months were 0.21, 0.24, 0.10, 0.15, and 0.09, respectively, and the maternal heritability of the corresponding traits was 0.12, 0.05, 0.04, 0.04, and 0.04, respectively. The genetic correlations between the body-weight traits were all positive and moderate-to-strong, except for the correlation between birth weight and the other body-weight traits. The significance of non-genetic factors studied in this work demanded a correction to improve the accuracy of the direct selection of lambs for body-weight traits. The estimated genetic parameters identified the weaning weight as a selection criterion for the improvement in body weight of Mecheri lambs at different ages. Inbred individuals accounted for approximately 13% of the total population in the Mecheri sheep population studied. There were 877 founders in the population, and the actual effective population size was 128.48. The population’s mean generation interval was 3.26. The mean inbreeding values ranged from 0.005 to 0.010 across generations. The population’s average relatedness ranged from 0.001 to 0.014 across generations. Individual inbreeding was found to be 0.45 per cent for the entire population and 3.4 per cent for the inbred population.

## 1. Introduction

Sheep farming in India contributes significantly to the livelihood of a large number of small and marginal farmers, as well as landless labourers. Mecheri sheep are one of the hair-type sheep breeds of India and are known for their relatively good growth rate, higher dressing percentage, and excellent skin quality. They are polled animals, medium-sized, and have a compact body covered with short hair. They are widely distributed in the southern state of India (i.e., Tamil Nadu), especially in the north-western agro-climatic zone, which includes Salem, Erode, and parts of the Dharmapuri and Namakkal districts [1]. Mecheri sheep are mainly reared for meat and play a crucial role in farmers’ economies by meeting their requirements. The tremendous demand for red meat in India makes sheep farming a profitable livestock enterprise. However, the insufficiency in the supply of sheep produce to meet the demand may limit the profitability of sheep enterprises. This insufficient production is primarily due to the slow growth rate of lambs [2]. Growth traits in farm animals, such as body weight, reflect the adaptability of the animals, and inefficiency in any of these interrelated traits significantly impacts the economy of sheep farming.

Identifying genetic and non-genetic sources of variation in economically important performance traits is a key component in designing animal-improvement programmes. The significance of maternal effects, in addition to direct genetic effects on production traits, had been studied in different sheep breeds as a part of the genetic evaluation programme. Disregarding maternal influence when it is essential skews the estimates of additive variance upwards [3,4]. Additionally, the identification of genetic correlations between economic traits may facilitate a simultaneous improvement of the correlated traits through indirect selection. Integration of the non-genetic factors and genetic parameters of the production traits of lambs may enable breeders to formulate optimal selection and management strategies for the improvement of the animals and better income generation.

The primary goal of population management is to maximize the long-term competitiveness of a breed. Hence, monitoring the rate of change in inbreeding and the genetic diversity within a population is critical for guiding breeding programmes. A breed’s conservation value could be enhanced by increasing its contribution to the gene pool of its species. This could include regaining its original genetic background and retaining a high level of genetic diversity. Hence, the breeding programme must ensure that the kinship does not increase quickly to maintain the desired effective size [5]. In a small population, the positive effects of making the breed genetically more similar may be outweighed by the negative effects of lost genetic diversity [6]. Hence, understanding the genetic structure of the breed and the rate of inbreeding accumulation is critical for developing appropriate management and breeding strategies that promote flock performance while maximizing genetic gain for future generations.

## 2. Materials and Methods

### 2.1. Data and Farm Management

Pedigree and growth details of Mecheri sheep maintained at Mecheri Sheep Research Station (MSRS), Tamil Nadu, India, were collected over a forty-year period from 1979 to 2018. The average annual rainfall and the mean maximum and minimum temperatures at this location were 894.38 mm, 34.30 °C, and 21.90 °C, respectively. At the MSRS, Mecheri sheep were allowed to graze for eight hours. When poor grazing was observed, the animals were fed with concentrates, the quantity of which was based on their age, sex, and production status. Depending on grazing conditions, each adult animal was supplemented with 200–350 g/d of concentrate. Breeding rams were fed 350–400 g of concentrate per day during the mating season. Aside from grazing, animals were also fed cultivated fodder and tree cuttings during the dry summer season. The breeding rams were maintained at a rate of 15 ewes per ram, and the mating of rams was limited to three to four consecutive breeding seasons. The number of ewes mated per rams in this study ranged between one and forty ewes, and the mating was conducted in such a way to avoid inbreeding by the careful verification of the pedigree structure. Controlled mating or hand mating was been practised to ensure the collection of pedigree data. All female lambs, excluding weak lambs, were selected and retained for breeding purposes. Lambing was limited to two seasons: the major season stretching from January to March (the main season), and the minor season from August to September (the off season). The maximum number of lambings always occurred during the main lambing season. Depending on their health condition, ewes were typically culled at around six to seven years of age.

### 2.2. Data Analysis

The growth performance of the Mecheri sheep was assessed for the body-weight traits recorded at birth (BW), three (3MW), six (6MW), nine (9MW), and 12 months (12MW) of age. After eliminating outliers, a data set on the growth information of 2616 lambs with pedigree details was created. The lambs were classified for non-genetic factors: namely, the sex of the lamb (male and female), season of birth (main and off), period of birth (1980 to 1984, 1985 to 1989, 1990 to 1994, 1995 to 1999, 2000–2004, 2005–2008, 2009–2013, and 2014–2018), type of birth (single and twin), and the parity of the ewe (parity—1, 2, 3, and 4 and above). All the non-genetic factors were taken as fixed effects, and the weight of the dam at lambing was included as a covariate in the statistical analysis. The initial analysis revealed that the interactions among the fixed effects were not significant and hence not considered in the final analysis [7]. A series of single-trait least squares analyses were carried out by fitting the following general linear model to assess the effects of different non-genetic factors on growth traits: Y_ijklmn_ = μ + P_i_ + C_j_ + D_k_ + S_l_ + B_m_+ e_ijklmn_
where Y_ijklmn_, μ, and e_ijklmn_ are the lamb’s body weight, overall mean for the trait, and the random error associated with each observation, respectively. The effects of the non-genetic factors, period of birth, season of birth, parity of dam, sex of lamb, and type of birth are mentioned in the model as P_i_, C_j_, D_k_, S_l_, and B_m_, respectively.

Significantly contributing non-genetic factors identified from the least squares analyses were included as fixed effects in animal models for the estimation of the (co)variance components of growth traits. By incorporating or excluding the maternal genetic and permanent environmental effects, with or without direct maternal genetic covariances in a simple random effect model, the following six different animal models [8] were fitted, using a restricted maximum likelihood for estimating various genetic parameters with the Wombat [9] software program.
y = Xb + Z_1_a+e (1)
y = Xb + Z_1_a + Z_2_m+e   Cov(a, m) = 0(2)
y = Xb + Z_1_a + Z_2_m+e    Cov(a, m) = Aσ_am_
(3)
y = Xb + Z_1_a + Z_2_pe+e (4)
y = Xb + Z_1_a + Z_2_m + Z_3_pe+e    Cov(a, m) = 0 (5)
y = Xb + Z_1_a + Z_2_m + Z_3_pe+e    Cov(a, m) = Aσ_am_
(6)
where y and e are the vectors of records (N × 1) and residual effects, respectively. In the animal models, fixed effects, additive genetic effects, maternal additive genetic effects, maternal permanent environmental effects, and residual effects are indicated as vectors b, a, m, and pe, respectively, and their corresponding incidence matrices are X, Z_1_, Z_2_, and Z_3,_ respectively. The following is the matrix of the assumed (co)variance structure:$$V\left(\begin{array}{c} \begin{array}{c} a\\ m\\ pe \end{array}\\ e \end{array}\right)=\left(\begin{array}{c} \begin{array}{c} A\sigma^{2}a\\ A\sigma \mathrm{am}\;\\ 0 \end{array}\\ 0 \end{array}\begin{array}{c} \begin{array}{c} A\sigma \mathrm{am}\\ A\sigma^{2}m\\ 0 \end{array}\\ 0 \end{array}\begin{array}{c} \begin{array}{c} 0\\ 0\\ I_{d}\sigma^{2}\mathrm{pe} \end{array}\\ 0 \end{array}\begin{array}{c} \begin{array}{c} 0\\ 0\\ 0 \end{array}\\ I_{n}n\sigma^{2}e \end{array}\right)$$

Direct additive and maternal genetic effects were assumed to be distributed normally with a mean of 0 and variances of Aσ^2^_a_ and Aσ^2^_m_, respectively. In the variance and covariance structure, A is the additive numerator relationship matrix, and σ^2^_a_, σ^2^_m_, and σ_am_ are the additive variances, maternal genetic variance, and the covariance between the direct and maternal genetic effects, respectively. Permanent environmental effects and residual effects were also assumed to follow a normal distribution with a mean of zero and variances of I_d_σ^2^_pe_ and I_n_σ^2^_e_, respectively. In the structure of the variance and covariance matrix, I_d_ and I_n_ are identity matrices with orders equal to the number of dams and individual records, respectively, and σ^2^_pe_ and σ^2^_e_ are the maternal permanent environment and residual variances, respectively. 

Heritability or variance ratio estimates were calculated as the ratio of the variance of additive genetic to the total phenotypic variance. Sampling errors of heritability and correlation estimates were derived as per the approximation method reported by Meyer [9]. A selection of the most appropriate animal model fitted was performed by using the likelihood ratio test [10,11]. When adding an effect caused the log likelihood to rise more than it did in the model in which the effect was not included, the added effect was considered to be a significant influencer. Differences in log likelihoods were compared to values for a chi-squared distribution with degrees of freedom equal to the difference in the number of covariance components fit for the two models. Statistical significance was determined at *p* < 0.05. Following this, the most-suited model for each trait according to the likelihood ratio test was employed in bivariate analyses to estimate the genetic, phenotypic, and environmental correlations between the studied traits. The variance and covariance components estimated from the best model in the univariate analysis were utilized as starting values in the bivariate analysis. The direct heritability (h^2^_a_ = σ^2^_a_/σ^2^_p_), maternal heritability (h^2^_m_ = σ^2^_m_/σ^2^_p_), genetic correlation between the direct and maternal effects (*r*_*a*,*m*_ = σ_*a*,*m*_(σ_*ax*_σ_*m*_), and the maternal permanent environmental variance (c^2^ = σ^2^_c_/σ^2^_p_) were estimated. The direct maternal correlation (r_am_) was computed as the ratio of the estimates of the direct maternal covariance (σ_am_) to the product of the square roots of the estimates of σ^2^_a_ and σ^2^_m_. For the purpose of estimating the inbreeding coefficient, a total of 4168 Mecheri sheep, including pedigreed individuals and their sires and dams, were used. The pedigree analysis was carried out using the Endog software, version 4.8 [12].

## 3. Results

Table 1 shows the details of the pedigree of the Mecheri sheep. A total of 2616 lambs were born from 226 rams and 1044 rams. The average number of records per ewe ranged from 1.59 (12MW) to 2.50 (BW), while rams had values ranging from 6.36 (12MW) to 11.57 (BW). The pedigree constitution check revealed enough information to estimate genetic components.

### 3.1. Growth Performance and Influence of Non-Genetic Factors

The mean values of the body weight of Mecheri sheep at birth, three, six, nine, and twelve months were 2.35, 9.76, 13.72, 16.68, and 19.46 kg, respectively. On least-square analysis, the influence of the period of birth on all the body-weight traits; sex on the 9MW and 12MW; birth type on birth weight, parity on the BW and 6MW; and the season of birth on the 3MW and 12MW (Table 2 and Figure 1a,b) were found to be significant (*p* < 0.05). 

### 3.2. (Co)variance Components and Heritability

(Co)variance components of the growth traits and heritability estimates pertaining to additive genetic, maternal genetic, and permanent environmental effects estimated for the Mecheri sheep are presented in Table 3. The model, which included direct and maternal genetic effects as random effects with no covariance between them (Model-2), was identified as the best-fitting model for explaining the (co)variance components. The direct additive heritability estimated from the best model for the BW, 3MW, 6MW, 9MW, and 12MW were 0.21, 0.24, 0.10, 0.15, and 0.09, respectively. Among the traits studied, comparatively higher heritability estimates were observed for the 3MW and the birth weight. The estimates of heritability for maternal genetic effects of the BW, 3MW, 6MW, 9MW, and 12MW were 0.12, 0.05, 0.04, 0.04, and 0.04, respectively. 

### 3.3. Genetic Correlation

The genetic correlations estimated between the body-weight traits were all positive (Table 4). Birth weight demonstrated a significant genetic correlation with weaning weight (0.66); however, it had a low correlation with other body-weight traits, which ranged from 0.19 to 0.28. The 3MW had a high genetic correlation with the 6MW (0.73) and 9MW (0.60) and a moderate correlation with the 12MW (0.53). However, the estimates of the genetic correlations of the 6MW and 9MW with succeeding body-weight traits were very high (0.82). In this study, birth weight had low phenotypic correlations with the other body-weight traits, whereas weaning and post-weaning traits had a strong correlation with body weight at later ages. 

### 3.4. Pedigree Analysis and Inbreeding 

The descriptive statistics and parameters describing the probability of gene origin in the Mecheri sheep population are presented in Table 5. The pedigree completeness level recorded for different generations ranged from 78.95 per cent in the first generation to 0.00001 per cent in the twelfth generation (Figure 2 and Table 5). The population of Mecheri sheep had an average equivalent generation of 1.74. The effective number of founder animals (f_e_) and effective number of ancestors (f_a_) for the pedigreed population were determined to be 117 and 99, respectively. It was predicted that the founders’ uneven contributions would increase inbreeding by 0.27 per cent. A significant subset of founders accounted for 50% of the population’s genetic diversity. The ratio of the effective number of founder animals (f_e_) to the effective number of ancestor animals (f_a_) was 1.18., and the mean value of the genetic conservation index (GCI) was 3.001. The mean generation interval (GI) for the gametic pathways, viz., sire–ram lamb, sire–ewe lamb, dam–ram lamb and dam–ewe lamb, were 3.02, 3.04, 3.59, and 3.45 years, respectively. The average GI for rams and ewes was 3.03 and 3.52 years, respectively, and was 3.26 years for the entire population.

The average relatedness and inbreeding coefficient were determined for each individual; their mean values for each generation are shown in Table 6. The average percentages of inbreeding for the whole population, pedigreed population, and inbred animals were 0.45, 0.61, and 3.41 per cent, respectively. In the whole population, only 1.65 per cent of the animals had more than 10 per cent inbreeding. The mean inbreeding values in the population ranged from 0.005 to 0.010 over generations. The generation-wise inbreeding increased from the second generation (0.5%) to subsequent generations and reached its maximum in the fifth generation (1.00%). Thereafter, it decreased and fluctuated between 0.86 per cent and 0.44 per cent across the sixth to twelfth generations (Figure 3). A total of 13.00 per cent of the Mecheri sheep population was inbred. The range of the mean average relatedness was 0.001 to 0.014 across generations. The average relatedness coefficient between individuals was estimated to be 0.89 and 1.1 per cent for the entire population and the pedigreed population, respectively.

## 4. Discussion

### 4.1. Growth Performance and Influence of Non-Genetic Factors

The body-weight traits obtained using the least squares mean at various ages were consistent with breed standards and earlier findings [13,14]. However, comparatively higher mean values of body-weight traits were reported earlier in the meat-type sheep breeds of India [15,16,17,18] and the Tazegzawt, Bonga, and Blackbelly sheep breeds in Africa [19,20,21]. Mecheri lambs grew by 70.50 percent in the first six months, with the remaining 29.50 percent of growth occurring in the following six months. The highest growth in the breed took place in the pre-weaning stage and decreased in the post-weaning stages.

Understanding non-genetic sources of variation in the performance traits of sheep is critical for developing the most effective management strategies to support the genetic progress of the animals. The non-genetic factor, viz., the period of birth, had a significant effect on all the body-weight traits but it did not show any apparent trend. The variation in body weight among lambs over periods may be attributable to differences in management, ram selection, and environmental variables such as ambient temperature, humidity, and rainfall [22]. A similar influence of the period of birth was reported in Moroccan Sardi sheep [7] and many other Indian meat-type sheep breeds [14,18,23,24]. The influence of season, expressed as a higher body weight in off-season-born lambs, could be attributed to the fact that the resource availability per lamb was greater in the off season than in the main season due to the large number of lambs born during the main season. Seasonal fluctuations in climate were also reflected in the differences in body weight over the course of the year. In many prior studies, season was found to be an influence for most of the body-weight traits in different Indian, hair-type meat breeds of sheep [14,24,25]. The body weight of the ram lambs at nine and twelve months of age was substantially superior to the ewe lambs. The growing differences in the endocrine systems of the male and female lambs may explain the discrepancies in body weight between the ram and ewe lambs as they grew older [26]. A similar effect of the sex of lambs on body weight at later ages were reported in different sheep breeds [14,18,25,27].

The parity of the dam significantly influenced the birth weight and six-months body weight. The body weight of lambs increased with the parity of their dam;therefore, the lambs born at a parity of third and above had significantly heavier bodies at birth. This could be due to the relative nutritional rivalry between the still-growing pregnant ewes and the growing fetus, as well as the fact that the uterine space and other factors necessary for lamb growth improve as the age of the dam increases [18,22]. Parity accounts for the maternal effects that often disappear as the lamb grows older. This is the reason why, after six months, no significant impact of parity was observed in the Mecheri lambs. Similar results were reported in Nellore [24] and Avikalin lambs [28]. Twin-born Mecheri lambs had significantly lower birth weights than singletons. Similar results were obtained in Balouchi [9] and Tazegzawt sheep [19]. The importance of the body weight of the dam at lambing revealed in this study was similar to earlier results in other sheep breeds [24,28,29,30].

### 4.2. (Co)variance Components and Heritability

The direct heritability (h^2^) estimates of the BW, 3MW, and 6MW were moderate, whereas the estimate was low for the 9MW. Lower heritability estimates for post-weaning traits in Mecheri sheep are consistent with the findings in Kermani [31], Kourdi [32], and Corriedale [33] sheep populations. However, Chokla [34], Avikalin [35], and Marwari [36] sheep all had higher estimates of direct heritability at post-weaning ages than Mecheri sheep. In comparison with a previous study on the same breed [14], the inclusion of maternal genetic effects showed a significant difference in direct heritability estimates. Many earlier studies stated that maternal effects, along with direct genetic effects, have a substantial effect on the body-weight traits of lambs in various breeds [10,37,38]. The heritability estimates derived for pre- and post-weaning growth traits in this study were comparable to those reported in other sheep breeds [22,24,25,34,39,40,41,42], but the pre-weaning estimates were superior to those of Egyptian Barki lambs [14]. 

In the current study and other studies, direct heritability decreased with increasing age [43,44,45,46,47]. The low heritability of post-weaning characteristics in Mecheri lambs proved that the majority of variance was caused by variables other than additive genetic factors. After weaning, the Mecheri lambs on the farm were allowed to graze as a small flock, exposing them to a range of environmental conditions and forage availability. Due to these farm practices, the environmental factors became a strong driver for post-weaning traits in the studied lambs. Therefore, the phenotypic selection based on these traits may not be favourable, and many generations of severe selection would be required to reap the benefits. Further, the findings suggested that to improve the post-weaning traits in Mecheri sheep, more focus should be placed on modifying the effects of non-genetic factors, such as improving environmental conditions and managerial and nutritional requirements. 

The maternal genetic influence was at its maximum at birth in this investigation. The maternal heritability estimated in other sheep breeds of India showed a similarity with the findings of the present study [15,24,25,34], except for the study on Malpura, which had a relatively higher maternal heritability estimate [39]. The estimated maternal heritability values for birth weight and weaning weight were similar to those of Iranian Sangsari sheep [48]. within addition to the maternal genetic effect, other maternal influences, such as the environment provided by the dam and the feeding behaviour of lambs until weaning, also played an important role in the development of lambs. As the milk productivity of Mecheri sheep is insufficient to meet the needs of fast-growing lambs, the addition of concentrate was practised at two months of age. After two months of lambing, the milk production in ewes decreases rapidly, which can lead to lower levels of maternal heritability at weaning. As a result of the decrease in lamb dependency on the mother, the contribution of maternal influences in phenotypic variation is expected to diminish with increasing lamb age. As a result, the proportion of permanent maternal environmental variance to phenotypic variance decreased.

The presence of a high level of genetic variability at weaning in Mecheri sheep can be utilized to design a breeding programme to improve the breed. Estimates of additive genetic variance and heritability for body-weight traits in this study indicate that there is space for the further genetic improvement of animals for these traits.

### 4.3. Genetic Correlation

The estimate of correlation (genetic) between the BW and 3MW in Mecheri sheep was positive and high, and the estimates were lower than the reported values of 0.56 for Columbia [49], 0.43 for Corridale [33], and 0.54 for Red Maasai [50] breeds of sheep. The strong genetic correlation between the BW and 3MW suggests that similar genetic and physiological mechanisms govern both variables. Selection for BW can result in significant improvements in the 3MW, modest improvements in marketing weights (6MW and 9MW), and a slight improvement in the 12MW. On the other hand, direct selection for a greater BW may result in more difficult births. Additionally, the trait is also influenced by maternal effects that must be considered.

The weaning weight in Mecheri sheep had strong genetic correlations with the 6MW and 9MW and a moderate correlation with the 12MW. The moderate heritability and the strong genetic correlations observed for the weaning weight suggest that the selection for the trait could result in an improved response in later body-weight traits. The genetic correlation observed in Mecheri sheep between the 3MW and 6MW was similar to the correlation found in Iranian Kordi sheep (0.73) [51], but it exceeded the values reported in Doyogena (0.35) [47] and Lori (0.52) [52] sheep. For Corriedale sheep, the value (0.913) reported was much higher [33]. Due to the strong genetic link between weaning and post-weaning variables in Mecheri sheep, it is desirable to record the 3 MW and deploy it for post-weaning body weight improvement programmes. As a result, time and money spent on data collection and genetic analysis will be reduced, enabling the breeding programme to progress faster for genetic improvement.

The genetic correlation between sequential traits was greater than the genetic correlation between non-consecutive traits, as predicted, and the observed trend is consistent with estimates from previous studies [33,50,53,54]. The estimates of maternal genetic correlation for various traits were positive and slightly higher than the values reported for Mehraban sheep [55]. In all growth traits studied, the maternal correlation was lower than the direct additive genetic correlation. The contribution of maternal influences on phenotypic variation is expected to diminish as the lambs age, and thus the ratio of permanent maternal environmental variance to phenotypic variance decreases. No antagonistic genetic correlations were observed between the body-weight traits, suggesting that the selection for any of these body weights will result in positive responses in other traits. 

Genotypic and phenotypic correlations between body-weight traits had a similar trend in that the estimates decreased as the age interval between weight measurements increased. The observed trend is consistent with estimates from previous studies [33,50,53,54]. The estimates of maternal genetic correlation for various traits were positive and slightly higher than the values reported for Mehraban sheep [55].

### 4.4. Pedigree Analysis and Inbreeding 

The average complete equivalent generation and percent pedigree completeness provide information about the pedigree depth of a population. The average equivalent generation recorded for the population of Mecheri sheep (1.74) was lower than the values reported for Nellore [18], Sandyno [56], Santa Ines [57], and Bharat Merino sheep [58]. Pedigree completeness up to the best-known generations is critical for reliable inbreeding estimates [59]. In the Mecheri sheep population, up to five generations (more than 5 per cent of ancestors) were known; however, after the fifth generation, comparatively few animals had known ancestors, indicating the difficulties in obtaining pedigrees. In Nilagiri, Sandyno, and Bharat Merino sheep, the per cent pedigree completeness for the first and fifth parental generations were 88 and 36 [60]; 81 and 21 [60]; and 91 and 58 [59], respectively. 

In the Mecheri population, only a few males were used as sires; hence, the effective number of founders (117) was less than the total number of founders (877). The reduced number of founder individuals contributed to a reduction in the effective population size and associated genetic diversity to a certain extent. This indicates that incorporating the effective number of founders in a pedigree report could give important information for the better management of inbreeding [59]. Comparatively, much lower effective numbers of founders were noted in Afshari [61] and Bharat Merino [58] sheep populations. 

The ratio of the effective number of founder animals to the effective number of ancestor (f_e_/f_a_) has been used to measure the degree of the genetic bottleneck in populations; and the ideal ratio would be one. It was stated that the marginal contribution from all ancestors must add up to one, and the f_a_ must always be smaller than or equal to the f_e_ [57]. The observed f_e_/f_a_ ratio in the Mecheri sheep population (1.18) demonstrated that unbalancing between ancestors and founders reduced the genetic variation to a minor degree and confirmed the absence of severe bottlenecks. This f_e_/f_a_ ratio is similar to the estimate in Afshari sheep (1.17) [61] but lesser than the estimates in Santa Ines (1.35) [57], Nilagiri (1.41) [56], and Bharat Merino (1.37) [59] sheep breeds. 

The projected rise in inbreeding (0.27) as a result of the unbalanced contribution of founders in Mecheri sheep was significant, but the number of ancestors contributing to 50 per cent of the population (42) was relatively high. Thus, the genetic differences from the founders had not been lost as a result of bottlenecks, unequal contribution, or segregation. Individuals in the population had genetic conservation index (GCI) values between 0.879 and 13.06. To maintain a balanced contribution from the founders, breeding rams could be selected from the animals with a higher index [62].

The estimated realized effective population size (128.48) was slightly higher than the reported estimates for Zandi (71) [61], Bharat Merino (89.29) [59], and Nilagiri (106.82) [56] sheep populations. The 50/500 rule [63] states that a population with an Ne of less than 50 is at risk of inbreeding depression, and a population with a minimum Ne of 500 is required to prevent the loss of genetic variance over centuries. According to Meuwissen [64], the critical Ne range is 50 to 100. Based on the above rules, it was confirmed that the estimated effective population size of the population under investigation is greater than the critical value. Although the Mecheri sheep population is not at risk from the immediate impacts of inbreeding, it is threatened by the loss of adaptive genetic variation in the long term.

Generation intervals in sire-based pathways (3.02 and 3.03 years) were comparatively lesser than the dam-based pathways (3.59 and 3.45 years). This could be attributed to the long-standing practise of keeping breeding ewes in a flock longer than breeding rams in order to produce offspring [65]. In the Mecheri flock, the retention of breeding ewes until they were six years old and the replacement of breeding rams after two to three breeding seasons were practised routinely. The average generation interval in the Mecheri flock of this study was comparable to the values reported in the Nilagiri (3.29 years) [56] and Nellore (3.37 years) [18] sheep populations.

The estimated inbreeding in the Mecheri sheep population (0.45%) was comparable to the estimates reported in the Iranian Moghani (0.40%) [66], Spanish Segurena (0.60%) [67], and Indian Nellore (0.97%) [18] sheep breeds. However, the estimate was much lower than the value estimated for Bharat Merino (2.36%) [58] sheep. Certain generations in the Mecheri population had lower levels of inbreeding than prior generations with respect to average inbreeding across generations. This decline could be attributed to the introduction of outside animals as part of breeding strategies geared towards Mecheri sheep improvement during those times. Very few animals in the studied population showed signs of inbreeding to a moderately high degree, reflecting the intensive use of a few sires. The highest level of inbreeding recorded in an individual (31.25%) in the studied population was similar to the value recorded in Nilagiri sheep (33.59%) [56]. However, a comparatively lower value was reported for the highest level of inbreeding in Nellore sheep [18]. Overall, the level of inbreeding in Mecheri sheep was substantially below the critical level as a result of the effective design and execution of a breeding plan that minimized the number of mating occurrences between related animals.

The average relatedness (AR) coefficient in the Mecheri sheep across generations showed a similar pattern to inbreeding, reaching its maximum in the fifth generation (1.4%). The average AR coefficient (0.89%) was notably greater than the mean inbreeding, which could be attributed to the comparatively good pedigree depth of the population studied. A comparable average AR (0.73%) was reported in Santa Ines sheep [57]. Comparatively higher percentages of average relatedness were reported in Nilagiri (3.45%) [56], Lori-Bakhtiari (2.27%) [68], and Bharat Merino (4.53%) [58] sheep breeds. A higher average relatedness combined with lower inbreeding is an indication of a high degree of relatedness among all members of the pedigree lineage.

## 5. Conclusions

The study revealed a prominent maternal influence in the pre-weaning growth trait and indicated the importance of maternal genes in determining this trait. The moderate heritability and the moderate-to-strong genetic correlations observed for the trait, weaning weight, suggest that this trait could be used as a selection criterion for improving the later-age growth traits. The analysis of pedigree information and genetic differences in the population revealed a high level of intra-flock diversity, with the effective population size exceeding the critical point, indicating sufficient genetic diversity for breed conservation. The rate of inbreeding in the population was under control, and the introduction of genetically superior rams from other populations to reduce inbreeding could be balanced by the flock’s selection and mating of superior breeding animals with a high genetic conservation index, which may maintain the founders’ balanced contribution.

## Figures and Tables

**Figure 1 animals-13-00454-f001:**
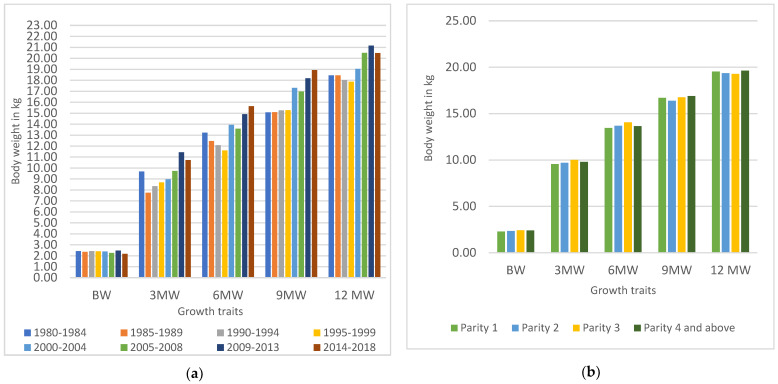
Influence of non-genetic factors on body-weight traits of Mecheri sheep: (**a**) effect of period of birth and (**b**) effect of parity.

**Figure 2 animals-13-00454-f002:**
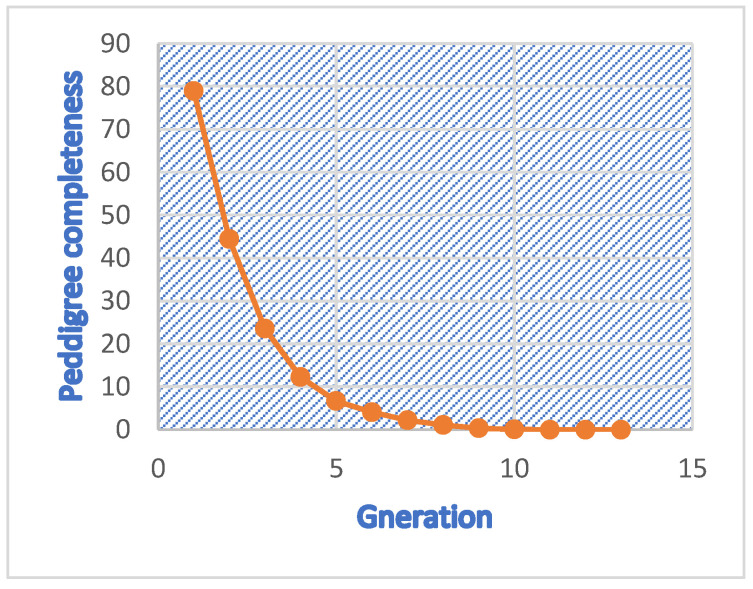
Per cent pedigree completeness in Mecheri Sheep.

**Figure 3 animals-13-00454-f003:**
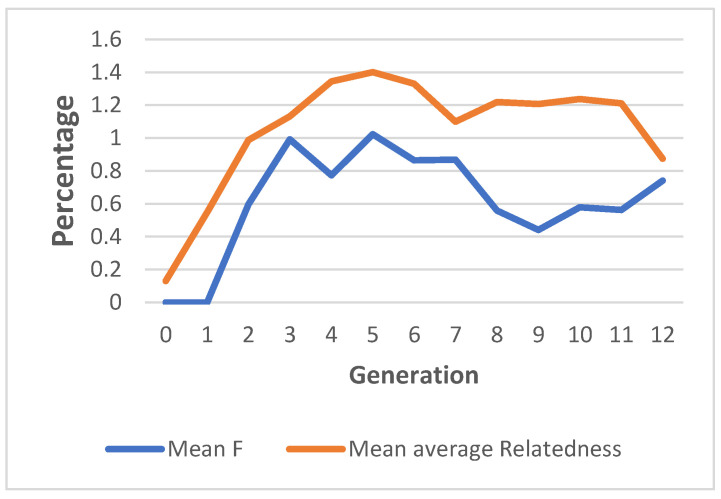
Inbreeding and average relatedness in Mecheri Sheep.

**Table 1 animals-13-00454-t001:** Data structure on growth characteristics.

Trait	Total Records	Number of Ewes	Number of Rams	Mean Records Per	
				Ewe	Ram
Birth	2616	1044	226	2.50	11.57
Weaning	2286	961	208	2.37	10.99
Six months	1578	814	183	1.93	8.62
Nine months	1203	701	167	1.71	7.20
One year old	1019	638	160	1.59	6.36

**Table 2 animals-13-00454-t002:** Least squares means (±SE) for body weight (kg) traits in Mecheri sheep.

Effect	BW	3MW	6MW	9MW	12MW
Overall	2.35 ± 0.11 (2616)	9.76 ± 0.06 (2286)	13.72 ± 0.10 (1578)	16.68 ± 0.11 (1203)	19.46 ± 0.12 (1019)
Sex				*	**
Male	2.38 ± 0.01 (1410)	9.90 ± 0.08 (1212)	14.16 ± 0.14 (682)	17.48 ^b^ ± 0.17 (469)	20.82 ^b^ ± 0.20 (352)
Female	2.32 ± 0.01 (1206)	9.62 ± 0.09 (1074)	13.30 ± 0.13 (896)	15.92 ^a^ ± 0.14 (734)	18.18 ^a^ ± 0.15 (667)
Period	**	**	**	**	**
1980–1984	2.42 ^a^ ± 0.02 (286)	9.68 ^c^ ± 0.13 (255)	13.22 ^bc^ ± 0.21 (172)	15.07 ^a^ ± 0.25 (147)	18.45 ^b^ ± 0.30 (123)
1985–1989	2.35 ^c^ ± 0.04 (195)	7.75 ^a^ ± 0.25 (141)	12.47 ^ab^ ± 0.39 (104)	15.09 ^a^ ± 0.43 (77)	18.45 ^a^ ± 0.44 (75)
1990–1994	2.43 ^b^ ± 0.03 (179)	8.34 ^b^ ± 0.14 (156)	12.08 ^a^ ± 0.19 (134)	15.25 ^a^ ± 0.22 (120)	18.02 ^ab^ ± 0.25 (97)
1995–1999	2.41 ^bd^ ±0.02 (167)	8.69 ^ab^ ± 0.22 (151)	11.61 ^a^ ± 0.30 (127)	15.28 ^a^ ± 0.24 (100)	17.87 ^a^ ± 0.29 (74)
2000–2004	2.39 ^cd^ ±0.02 (312)	8.98 ^b^ ± 0.16 (266)	13.93 ^b^ ± 0.22 (253)	17.31 ^b^ ± 0.28 (196)	19.04 ^ab^ ± 0.33 (158)
2005–2008	2.26 ^b^ ± 0.02 (635)	9.74 ^b^ ± 0.15 (596)	13.59 ^c^ ± 0.20 (449)	16.98 ^b^ ± 0.23 (389)	20.50 ^c^ ± 0.24 (345)
2009–2013	2.47 ^e^ ± 0.02 (585)	11.44 ^e^ ± 0.15 (558)	14.91 ^d^ ± 0.25 (300)	18.18 ^c^ ± 0.36 (146)	21.16 ^d^ ± 0.39 (127)
2014–2018	2.17 ^ac^ ± 0.03 (257)	10.72 ^e^ ± 0.22 (163)	15.64 ^d^ ± 0.44 (39)	18.93 ^c^ ± 0.51 (28)	20.48 ^cd^ ± 0.58 (20)
Season		**			*
Main	2.36 ± 0.01 (1810)	9.09 ^a^ ± 0.06 (1572)	13.20 ± 0.09 (1161)	16.48 ± 0.12 (899)	18.98 ^a^ ± 0.13 (750)
Off	2.35 ± 0.02 (806)	10.60 ^b^ ± 0.12 (714)	14.44 ± 0.19 (417)	17.03 ± 0.22 (304)	20.24 ^b^ ± 0.25 (269)
Birth type	**				
Single	2.42 ^b^± 0.09 (2577)	9.60 ± 0.05 (2257)	13.67 ± 0.09 (1569)	16.63 ± 0.11 (1200)	19.39 ± 0.12 (1016)
Twin	1.98 ^a^ ± 0.04 (39)	10.86 ± 0.34 (29)	14.46 ± 0.76 (9)	18.26 ± 1.30 (3)	21.57 ± 1.37 (3)
Parity	*		*		
One	2.28 ^a^ ± 0.02 (948)	9.56 ± 0.11 (786)	13.46 ^a^ ± 0.17 (564)	16.70 ± 0.19 (435)	19.54 ± 0.24 (351)
Two	2.34 ^b^ ± 0.02 (685)	9.69 ± 0.13 (594)	13.69 ^c^ ± 0.17 (405)	16.39 ± 0.22 (311)	19.37 ± 0.24 (268)
Three	2.42 ^c^ ± 0.01 (473)	10.01 ± 0.12 (440)	14.07 ^b^ ± 0.21 (329)	16.76 ± 0.23 (266)	19.28 ± 0.24 (232)
Four and above	2.40 ^c^ ± 0.02 (510)	9.80 ± 0.15 (466)	13.66 ^b^ ± 0.23 (280)	16.9 ± 0.26 (191)	19.64 ± 0.28 (168)
BWD	**	**	**	**	**

BWD—Body weight of dam at lambing; ** *p* < 0.01; * *p* < 0.05; figure in parentheses indicates number of observations. Means bearing same superscript do not differ significantly.

**Table 3 animals-13-00454-t003:** (Co)variance components and heritability estimates of body-weight traits.

Model	Component	BW	3MW	6MW	9MW	12MW
Model 1	σ^2^_a_	0.08	0.8	0.69	0.95	0.88
	σ^2^_e_	0.25	2.14	5.67	6.01	7.77
	σ^2^_p_	0.33	2.94	6.36	6.96	8.65
	h^2^	0.24 ± 0.04	0.27 ± 0.04	0.11± 0.03	0.14 ± 0.04	0.10 ± 0.06
Model 2 *	σ^2^_a_	0.07	0.72	0.56	0.98	0.78
	σ^2^_m_	0.04	0.14	0.24	0.24	0.34
	σ^2^_e_	0.23	2.12	4.98	5.25	7.53
	σ^2^_p_	0.34	2.98	5.78	6.47	8.65
	h^2^	0.21 ± 0.05	0.24 ± 0.05	0.10 ± 0.04	0.15 ± 0.04	0.09 ± 0.06
	m^2^	0.12 ± 0.05	0.05 ± 0.02	0.04 ± 0.01	0.04 ± 0.01	0.04 ± 0.02
Model 3	σ^2^_a_	0.04	0.84	0.93	0.98	0.92
	σ^2^_m_	0.03	0.23	0.24	0.25	0.26
	σ_am_	−0.85	−1.42	−2.27	−3.32	−3.28
	σ^2^_e_	1.04	3.78	7.36	8.68	10.59
	σ^2^_p_	0.26	3.43	6.26	6.59	8.49
	h^2^	0.15 ± 0.05	0.24 ± 0.07	0.15 ± 0.05	0.15 ± 0.04	0.11 ± 0.05
	m^2^	0.12 ± 0.05	0.07 ± 0.02	0.04 ± 0.02	0.04 ± 0.01	0.03 ± 0.01
	r_am_	−0.987	−0.993	−0.993	−0.967	−0.957
Model 4	σ^2^_a_	0.082	0.86	0.87	0.98	0.79
	σ^2^_c_	0.043	0.29	0.32	0.28	0.34
	σ^2^_e_	0.243	2.154	5.23	5.84	7.14
	σ^2^_p_	0.368	3.304	6.42	7.1	8.27
	h^2^	0.22 ± 0.05	0.26 ± 0.07	0.14 ± 0.03	0.14 ± 0.06	0.10 ± 0.05
	c^2^	0.117	0.088	0.049	0.039	0.041
Model 5	σ^2^_a_	0.08	0.85	0.67	0.943	0.824
	σ^2^_m_	0.06	0.189	0.231	0.234	0.143
	σ^2^_c_	0.02	0.167	0.28	0.27	0.18
	σ^2^_e_	0.212	2.234	5.46	5.83	7.85
	σ^2^_p_	0.372	3.44	6.641	7.277	8.997
	h^2^	0.22 ± 0.06	0.25 ± 0.04	0.10 ± 0.06	0.13 ± 0.04	0.00 ± 0.05
	m^2^	0.16 ± 0.04	0.05 ± 0.02	0.03 ± 0.01	0.03 ± 0.02	0.02 ± 0.01
	c^2^	0.053	0.048	0.042	0.037	0.020
Model 6	σ^2^_a_	0.06	0.734	0.598	0.854	0.814
	σ^2^_m_	0.03	0.189	0.183	0.284	0.283
	σ_am_	−0.68	−2.34	−2.85	−3.12	−3.28
	σ^2^_c_	0.024	0.227	0.267	0.252	0.34
	σ^2^_e_	0.945	4.234	8.56	7.89	9.89
	σ^2^_p_	0.379	3.044	6.758	6.16	8.047
	h^2^	0.16 ± 0.06	0.24 ± 0.04	0.09 ± 0.06	0.14 ± 0.04	0.10 ± 0.05
	m^2^	0.08 ± 0.04	0.06 ± 0.02	0.03 ± 0.01	0.05 ± 0.02	0.04 ± 0.01
	r_am_	−0.967	−0.987	−0.998	−0.978	−0.979
	c^2^	0.063	0.074	0.039	0.040	0.042

σ^2^_e_—error variance; σ^2^_p_—phenotypic variance; h^2^—direct heritability; c^2^—maternal permanent environmental variance ratio; m^2^—maternal heritability; σ_am_—direct-maternal genetic covariance; r_am_—direct- maternal genetic correlation; * best-fitting model.

**Table 4 animals-13-00454-t004:** Genetic and phenotypic correlations among body-weight traits in Mecheri sheep.

Trait	BW	3MW	6MW	9MW	12MW
BW		0.34 ± 0.12	0.26 ± 0.11	0.24 ± 0.20	0.20 ± 0.21
3MW	0.66 ± 0.11		0.64 ± 0.12	0.46 ± 0.14	0.45 ± 0.16
6MW	0.28 ± 0.11	0.73 ± 0.15		0.80 ± 0.07	0.70 ± 0.12
9MW	0.27 ± 0.21	0.60 ± 0.17	0.94 ± 0.05		0.81 ± 0.06
12MW	0.19 ± 0.23	0.53 ± 0.19	0.82 ± 0.11	0.91 ± 0.07	

Genetic correlations are below diagonal and phenotypic correlations are above diagonal.

**Table 5 animals-13-00454-t005:** Population parameters describing probability of gene origin in Mecheri sheep.

Population Parameter	Value/Estimate
Population size	4168
Reference (pedigreed) population size	3063
Actual base population size (Assuming one unknown parent = half founder)	877
Effective number of founder animals in the reference population (f_e_)	117
Effective population size of founder	182.83
Number of ancestors contributing to the reference population	648
Effective number of ancestor animals for the reference population (f_a_)	99
f_e_/f_a_ ratio	1.18
Number of ancestors explaining 50% variability	42
Realized effective population size (Ne)	128.48
Mean complete generation	1.00
Mean equivalent generation	1.74
Proportion of inbred individuals in the population	0.1312
Proportion of inbred individuals in the population	0.1312

**Table 6 animals-13-00454-t006:** Mean inbreeding (F), average relatedness, and per cent pedigree completeness.

Generation	Animals (N)	Mean F	% Inbred Individuals	Average F for Inbred Individuals	Mean Average Relatedness	% Pedigree Completeness
0	649	0	0	0	0.0013	0.00
1	854	0	0	0	0.0054	78.95
2	460	0.0059	0.0282	0.2115	0.0098	44.42
3	359	0.0099	0.0863	0.1149	0.0113	23.57
4	213	0.0077	0.1361	0.0568	0.0134	12.28
5	209	0.0102	0.2344	0.0436	0.0140	6.66
6	166	0.0086	0.2228	0.0387	0.0133	4.03
7	159	0.0086	0.1194	0.0727	0.0109	2.24
8	206	0.0055	0.2184	0.0256	0.0122	1.05
9	391	0.0044	0.2429	0.0181	0.0120	0.31
10	292	0.0057	0.4246	0.0136	0.0123	0.06
11	179	0.0056	0.5195	0.0108	0.0121	0.00
12	28	0.0074	0.4285	0.0172	0.0087	0.00
13	3	0	0	0	0.0088	0.00

## Data Availability

The data set created and analysed in the current study will be made available on reasonable request.

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
