# Peer review of "Genetic Parameters of Growth Traits and Quantitative Genetic Metrics for Selection and Conservation of Mecheri Sheep of Tamil Nadu"

_animals, 2023, doi:10.3390/ani13030454_

Round 1
Reviewer 1 Report
Dear Authors
The work is interesting, but need improuvement especialy methodologie (describe more heritability estimation methode) and discution (the majority of your compareaon where with Indian works).
Follow my remarks as mentioned in the attached file.
Best regards

Author Response
Dear Sir
Thank you for reviewing my paper and the following corrections have been made as per your suggestion
Kr
AKT

Reviewer 2 Report
Please look at the attached files

Author Response
Dear sir,
Thank you for reviewing my paper and the following corrections have been made as per your suggestion.
KR
AKT

Reviewer 3 Report
It is advisable to give a little more background on Mecheri sheep.
Author Response

(The authors gave the same response as above.)

Round 2
Reviewer 2 Report
Explanation for the study was much clear and improved.
However, minor spell check was still needed regarding to language and tyle.
Author Response
Dear sir,
Thank you for reviewing our paper and the corrections have been made as per your suggestions and thank you for your valuable suggestions
KR
AKT
